# Validating Immunomodulatory Responses of r-*Ld*ODC Protein and Its Derived HLA-DRB1 Restricted Epitopes against Visceral Leishmaniasis in BALB/c Mice

**DOI:** 10.3390/pathogens12010016

**Published:** 2022-12-22

**Authors:** Rajkishor Pandey, Rohit Kumar Gautam, Simran Sharma, Mebrahtu G. Tedla, Vijay Mahantesh, Manas Ranjan Dikhit, Akhilesh Kumar, Krishna Pandey, Sanjiva Bimal

**Affiliations:** 1Department of Biotechnology, National Institute of Pharmaceutical Education & Research, Hajipur 844102, India; 2Division of Immunology, Rajendra Memorial Research Institute of Medical Sciences, Patna 800007, India; 3Department of Child Health, School of Medicine, University of Missouri-Columbia, Columbia, MO 65211, USA; 4Department of Basic and Applied Sciences, National Institute of Food Technology Entrepreneurship & Management (NIFTEM), Kundli, Sonipat 131028, India; 5Department of Biomedical Informatics, Rajendra Memorial Research Institute of Medical Sciences, Patna 800007, India; 6Department of Clinical Medicine, Rajendra Memorial Research Institute of Medical Sciences, Patna 800007, India

**Keywords:** *Leishmania donovani*, visceral leishmaniasis, ODC, T cell, MHC Class-II, reverse vaccinology

## Abstract

Vaccination is considered the most appropriate way to control visceral leishmaniasis (VL). With this background, the r-*Ld*ODC protein as well as its derived HLA-DRB1-restricted synthetic peptides (P1: RLMPSAHAI, P2: LLDQYQIHL, P3: GLYHSFNCI, P4: AVLEVLSAL, and P5: RLPASPAAL) were validated in BALB/c mice against visceral leishmaniasis. The study was initiated by immunization of the r-*Ld*ODC protein as well as its derived peptides cocktail with adjuvants (r-CD2 and MPL-A) in different mice groups, separately. Splenocytes isolated from the challenged and differentially immunized mice group exhibited significantly higher IFN-γ secretion, which was evidenced by the increase in the expression profile of intracellular CD4+IFN-γ T cells. However, the IL-10 secretion did not show a significant increase against the protein and peptide cocktail. Subsequently, the study confirmed the ability of peptides as immunoprophylactic agents, as the IE-I/AD-I molecule overexpressed on monocytes and macrophages of the challenged mice group. The parasitic load in macrophages of the protein and peptides cocktail immunized mice groups, and T cell proliferation rate, further established immunoprophylactic efficacy of the r-*Ld*ODC protein and peptide cocktail. This study suggests that the r-*Ld*ODC protein, as well as its derived HLA-DRB1-restricted synthetic peptides, have immunoprophylactic potential and can activate other immune cells’ functions towards protection against visceral leishmaniasis. However, a detailed study in a humanized mice model can explore its potential as a vaccine candidate.

## 1. Introduction

Visceral leishmaniasis (Kala-azar) is a parasitic disease caused by *L. donovani* and *L. infantum/L. chagasi* in the old world and new world, respectively. This parasite is transmitted between vertebrates through the bite of blood-sucking phlebotomine sand flies [1,2,3]. An estimation of more than 500,000 million VL cases and 50,000 deaths have been reported worldwide. It was also reported that people living in developing countries are more susceptible to *Leishmania* infection, which predominantly affects organs such as liver, spleen, and bone marrow of patients [4,5,6]. At present, AmBisome (a liposomal form of Amphotericin B) and Miltefosine are effective medicines for the treatment of VL. However, they have limitations due to host toxicity, developing resistances, route of administration, cost, etc. [7,8]. To overcome these limitations, vaccines could be an economical alternative approach for treating *Leishmania* infection. However, until now there is no cost-effective vaccine available and none of the vaccine candidates were successful in all the stages of the clinical trial against VL [9,10].

The *Leishmania* parasite is an intracellular parasite that undergoes division in host phagolysosomes [11]. The T cells are major immune cells involved in the elimination of parasites to suppress infection [12]. During exposure to *Leishmania* infection, macrophages engulf the internalized parasites and digest them [13], and the digested fragments of parasites are presented on the surface of macrophages or other antigen-presenting cells through collaborating with MHC molecules [14]. Fragmented peptides on the surface of antigen-presenting cells interact with T cells or newly activated macrophages for the generation of proinflammatory cytokines such as IFN-γ and IL-12 [12,15,16]. Apart from cytokine production, the antigen-stimulated macrophages increase the expression of MHC Class-II molecules [17]. It has been found that the addition of adjuvants with antigen has significantly improved the immunoprophylactic efficacy, and hence vaccine potential [16,18]. In an earlier study, polypeptide-based vaccines, Leish-IIIf with adjuvants GLA-SE and MPL-A demonstrated effective immune responses against the *Leishmania* parasite [19,20]. Therefore in the present study, adjuvant MPL-A and recombinant CD2 were used as immunomodulator/adjuvant to magnify Th1 protective responses against VL. CD-2 is a surface adhesion molecule that facilitates T cell’s adhesive interaction with other immune cells. In another study, the CD2 interacts as co-stimulatory molecules with CD28/CD58 surface receptors of T cells [21,22].

The development of an effective and safe vaccine against *Leishmania* requires identifying the molecules which elicit immune protection, so understanding the cellular and molecular pathway of immune regulation is critical. As mentioned in the literature, the polyamine biosynthesis pathway and its derived byproducts played a crucial role in the survival and proliferation of the *Leishmania* parasite [23,24,25]. An in vitro and in vivo experiment study reported that ODC protein, a rate-limiting enzyme of polyamine biosynthesis pathway [26], ODC DNA construct [27], as well as its derived synthetic peptides [16,28] have the potential to generate a significant amount of pro-inflammatory cytokines and promote protection against *Leishmania* infection. Another study reported HIV-1 reverse transcriptase (RT) with fused ornithine decarboxylase (ODC) as a potential vaccine candidate as it strengthened a strong Th1 immune response against HIV in mice models [29]. In the present study, the purified recombinant *Leishmania donovani* ornithine decarboxylase (r-*Ld*ODC) protein and its derived HLA-DRB1-restricted peptides were validated against VL in BALB/c mice model.

## 2. Materials and Methods

### 2.1. Mice and Parasites

In this study, six- to eight-week old male inbred BALB/c mice were obtained from the National Center for Laboratory Animals Science, Telangana, India. The study was conducted following the recommendation of the Institutional Animal Ethical Committee of the Rajendra Memorial Research Institute of Medical Science (RMRIMS), Bihar, India (RMRIMS/IAEC/06/2017-2018). Accordingly, the procured and inbred mice were grown in a hygienic, pathogen-free, air-cooled animal facility. The stationary phase cultured promastigote form of *Leishmania donovani* (Ag83) was used for all experiments. The promastigote form of the parasite was obtained after conversion of amastigote which was isolated from the spleen of an infected male Golden Syrian hamster (*Mesocricetus auratus*; 8–10 weeks old) and the cell was cultured at 24 °C in Schneider’s insect medium (Thermo Fisher Scientific, St. Louis, MO, USA through GIBCO Life Technology, Delhi, India), with 10% heat-inactivated fetal bovine serum (Gibco, Thermo Fisher Scientific, Waltham, MA, USA), 20 mM L-glutamine, 100 units/mL of penicillin, and 50 μg/mL of gentamycin (Thermo Fisher Scientific, Waltham, MA, USA) at pH 7.4.

### 2.2. Soluble Leishmania Antigen (SLA)

*Leishmania* soluble antigen was prepared as mentioned in the earlier protocol [30]. In brief, the stationary phase culture of *L. donovani* cells (1 × 10^8^/mL) was added with 5 mL of cold, sterile phosphate buffer saline (PBS), and then subjected to five freeze-thaw cycles involving freezing at −196 °C (in liquid nitrogen) and thawing at 37 °C in a water bath. Simultaneously, the lysed sample was centrifuged at 10,000× *g* for 20 min at 4 °C. The supernatant containing SLA was collected and stored at −80 °C until use. The protein concentration was measured using a BCA Kit (Thermo Fisher Scientific, MA, USA) according to manufactured protocol.

### 2.3. Protein and Peptide Synthesis

The r-*Ld*ODC protein was cloned, expressed, and purified according to the previous study [26]. In the recombinant protein, Bacterial lipopolysaccharide/endotoxin (LPS) impurity was checked by using the Limulus amoebocyte lysate (LAL) test (Thermo Fisher Scientific Nunc, MA, USA). Endotoxin was removed by passing r-*Ld*ODC protein through a polymyxin B-agarose column (Sigma-Aldrich, St. Louis, MO, USA) according to the manufacturer’s instructions. Endotoxin-free protein was used in further study.

Apart from recombinant protein, the five most potent immunogenic *Ld*ODC-derived HLA-DRB10201 (MHC Class-II) restricted 15 mer epitopes against *Homo sapiens* namely, YNVVTRLPASPAALA (P1), SERIRMAPPASASKA (P2), PGRYFTAASHALLMN (P3), ALLMNVFASRTLRLS (P4), and EKISRLMPSAHAIIR (P5) were identified by SYFPEITHI, IEDB, and NetMHCCII-2.2 bioinformatic tools. Further population coverage and antigen cross-presentation were analyzed and screened by the IEDB population coverage analysis tool and NetMHCpan 2.3, respectively. Additionally, a 100 ns molecular dynamics simulation study was performed for each selected epitope at 300 K. The shortlisted epitopes were synthesized by Peptide2.0 (Chantilly, VA, USA) with 95% purity [16]. The purified protein and synthesized peptides were stored at −80 °C for further in vivo study. 

### 2.4. Immunization of Mice

Immunization of female BALB/c mice was performed as per the previously described methodology [27,30]. In brief, a total of 30, six- to eight-week-old female BALB/c mice were kept under pathogen-free conditions. Six mice were reserved in each group to perform the experiments, these groups were (1) a healthy uninfected control group (Placebo group), (2) a group of mice immunized with Soluble Leishmania Antigen (SLA) (50 µM/mL) as a positive control, (3) a group of mice immunized with only adjuvants monophosphoryl lipid-A (MPL-A) (25 µM/mL, Sigma-Aldrich, St. Louis, MO, USA) and r-CD2 (4 µM/mL, Becton, Dickinson and Company, Franklin Lakes, NJ, USA), (4) a group of mice immunized with r-*Ld*ODC protein (50 µM/mL) with adjuvants MPL-A and r-CD2, and (5) a group of mice immunized with a cocktail of the peptides (50 µM/mL) with adjuvant MPL-A and r-CD2. Each group of mice was immunized subcutaneously (at both adjacent abdominal sites (50 µL each site)) with either normal saline, r-*Ld*ODC protein, or the cocktail of peptides. Similarly, SLA and normal saline (both at the concentration of 100 μL/mouse) were used as the positive and negative control, respectively (100 μL/mouse). One set of mice was immunized with only adjuvant (MPL-A and r-CD2) to check the immune responses of adjuvant in mice (100 μL/mouse). All immunogenic antigens were formulated with adjuvant MPL-A (25 µM/mL) and CD2 (4 µM/mL) with normal saline (0.85% NaCl). Each group of mice was subcutaneously immunized with formulated antigens with the addition of adjuvants as the first immunization. Two boost-up doses were given on the 7th and 15th day. (Each antigen was used at the concentration of 50 µM/mL in the first immunization and boost-up dose).

### 2.5. Splenic Mononuclear Cell Isolation

After 30 days of the second boost-up immunization, each group of mice was euthanized and dissected. The spleen was homogenized separately to isolate splenocytes. Then peripheral blood mononuclear cells (PBMCs) were isolated by using the Histopaque-1077 density gradient method (Sigma-Aldrich, Darmstadt, Germany). In brief, the isolated splenocyte of each mouse was diluted at a 1:1 ratio with sterile PBS separately, poured on a 3 mL histopaque-1077 containing tube, and centrifuged at 400× *g* for 30 min. The separated mononuclear layer was washed twice with sterile PBS (5 mL/wash), counted, and utilized within an hour for further experiments. PBMCs from isolated splenocytes (1 × 10^6^ cells/well) were cultured in six-well plates with the addition of RPMI 1640 medium containing 100 μg/mL penicillin, 100 μg/mL streptomycin, and heat inactivated 10% fetal bovine serum (Gibco, Thermo Fisher Scientific, MA, USA).

### 2.6. Quantification of Secretory IFN-γ in Protein and Peptides Cocktail Immunized Mice Group

The level of IFN-γ concentration was quantified using cultured supernatant of each group of immunized mice according to the manufacturer protocol (ELISA Kit, BD Bioscience, San Jose, CA, USA). Briefly, cultured mononuclear splenocytes (1 × 10^6^ cells/well) from different immunized mice groups were stimulated by r-*Ld*ODC protein and peptides cocktail (10 µg/mL) in a six-well plate and incubated for 48 h in the control condition (at 5% CO_2_ and 37 °C). The cultured cells’ supernatant was collected and centrifuged for quantification of IFN-γ concentration from each immunized mice group. In each assay, a standard curve was made by a known concentration of corresponding cytokine (BD Bioscience, CA, USA). Here, the SLA and normal saline immunized mice groups were used as the positive and negative control, respectively. The sensitivity of the enzyme-linked immunosorbent assay (ELISA) was 15 pg/mL for IFN-γ.

### 2.7. Monitoring the T Cell Proliferation in Different Immunized Mice Groups

The T cells (supernatant of cultured splenocytes cells, assuming T cells) were collected from the cultured supernatant of mononuclear cells of each immunized mice group and stained by carboxyfluorescein succinimidyl ester (CFSE) dye (2 µM/mL). The stained T cells were seeded back in the same well (6-well plate) of an immunized group and triggered with r-*Ld*ODC protein as well as peptides cocktail (10 µg/mL). Subsequently, the cultured cells were further incubated at 37 °C and 5% CO_2_ for 96 h. Then cultured cells were harvested, washed with stain buffer (1% FBS in PBS), and acquired on fluorescence-activated cell sorting (FACS) caliber for further analysis.

### 2.8. Measurement of Intracellular Cytokines Production and I-AD/I-ED Expression in Protein and Peptide Cocktail Immunized Mice

Mononuclear splenocyte cells of different immunized mice groups were isolated separately by Histopaque-1077 density gradient methods following manufacturer protocol (Sigma-Aldrich, USA). The seeded cells (1 × 10^6^/well) were treated with r-*Ld*ODC protein and peptides cocktail (10 µg/mL) and incubated at 5% CO_2_ and 37 °C for 24 h. During the last 4 h of incubation, brefeldin-A (1 µg/mL) was added to the cultured tube. The cultured cells were collected and washed twice with stain buffer (1% FBS in sterile PBS). Washed cells were stained with anti-mouse FITC-CD4 conjugated antibody (BD Bioscience, CA, USA) as a surface marker and incubated for 30 min at 4 °C. The cells were fixed and permeabilized with Cytofix-Cytoperm buffer (BD Bioscience, CA, USA). Permeabilized cells were stained with anti-mouse PerCp-IFN-γ and PE-IL-10 conjugated antibodies and incubated at 4 °C for 30 min. In parallel, isotype control of the corresponding cytokine was used in each experiment. Similarly, triggered splenocyte cells were harvested, washed, and stained with anti-mouse PE-CD14+ antibody, and incubated for 30 min at 4 °C. After staining, cells were fixed and permeabilized with Cytofix and Cytoperm buffer (BD Bioscience, CA, USA). Perm washed cells were stained with FITC conjugated anti-mouse I-AD/I-ED antibody and incubated at 4 °C for 30 min. Isotype control was used in parallel in each experiment. At least 30,000 events were obtained for the detection of percentage gated cytokine patterns on FACS caliber. The data were analyzed on FACS caliber and cellquest software provided by BD Bioscience.

### 2.9. Monitoring the Parasite Load in the Macrophage

The *Leishmania* parasite was collected from the cultured flask (T-25 cm^2^, Thermo Fisher Scientific Nunc, MA, USA), washed with sterile PBS, and stained by CFSE dye (1 µg, BioLegend, San Diego, CA, USA) according to the previously described method [27]. Splenocytes from each immunized mice group, pre-seeded on six-well plates (1 × 10^6^ cells/well) were challenged with the stained parasites in a 1:10 ratio (monocyte: parasite). The plate was incubated in a CO_2_ incubator for 48 h. The cultured cells were harvested separately and washed twice with stain buffer (1% FBS in PBS). Subsequently, the sample was acquired on flow cytometry and gated macrophages were used for monitoring the parasite proliferation rates.

### 2.10. Statistical Analysis

All analyzed data are expressed as the mean ± SEM (standard error of the mean). The significance was assessed by one-way analysis of variance (ANOVA) with Tukey’s post hoc multiple comparison tests. Each experiment was run in triplicate, and a value of *p* ≤ 0.05 was considered for significant. Statistical analysis was done by using Graph Pad Prism software version 5.0 (GraphPad Software Inc., San Diego, CA, USA).

## 3. Results

### 3.1. The r-LdODC Protein and Its Derived Synthetic Peptides Characterization

Endotoxin-free recombinant *Ld*ODC protein (10 µg/mL) has already been characterized in our previous study [26], where the top five best immunogenic *Ld*ODC-derived synthetic peptides (P1: RLMPSAHAI, P2: LLDQYQIHL, P3: GLYHSFNCI, P4: AVLEVLSAL, and P5: RLPASPAAL) showed 100% population coverage. The conservancy of selected peptides in different species of *Leishmania* was predicted between 46.62 and 100%. Further, the molecular dynamics simulation study revealed stable interaction and compact conformation between peptides and HLA molecules on the 100 ns time scale. Additionally, the in vitro screening of peptides showed significant production of proinflammatory cytokines (CD4+IFN-γ and CD14+IL-12) in the treated cases of VL subjects. However, the production of anti-inflammatory cytokine CD4+IL-10 was not significantly higher (*p* ≥ 0.05) [16].

### 3.2. Quantification of IFN-γ by ELISA

IFN-γ is a major pro-inflammatory cytokine that inhibits the *Leishmania* parasite infection. It is secreted from various immune cells and kills the parasite directly. Results from the protein and peptide cocktail (10 µg/mL) stimulated PBMCs of the immunized mice group showed that the antigens group produced two-fold higher IFN-γ quantification as compared to the normal saline group (Figure 1). On contrary, the saline group and adjuvant (MPL-A and r-CD2) immunized group showed non-significant IFN-γ production. Parallelly, in each experiment, the SLA immunized mice group was used as a positive control.

### 3.3. Measurement of T Cell Proliferation

CFSE-labeled T cells were assessed for proliferative response in each immunized mice group. The recombinant *Ld*ODC protein and peptide cocktail immunized group showed a three-fold higher T cell proliferation rate (*p* ≤ 0.05) as compared to the placebo and adjuvant group. However, normal saline, as well as the adjuvant group, exhibited non-significant proliferation. Specifically, the protein immunized mice group presented a higher T cell proliferation rate than the cocktail of peptides. In each experiment, the SLA group was used as a positive control (Figure 2).

### 3.4. Assessment of Intracellular Cytokine Production against Protein and Peptides Cocktail

The present study further explored the measurement of inflammatory cytokine in the immunized mice group. It was noted that cytokines provoke the function of immune cells toward antigen responses. In this context, the results of the protein and peptide cocktail immunized group showed a significantly higher amount of CD4+IFN-γ production than the placebo group (*p* ≤ 0.05). While normal saline and the adjuvant immunized group showed no significant amount of IFN-γ production. In addition, the SLA immunized mice group showed a significantly higher amount of production when it was used as a positive control (Figure 3A,C). In contrast, the CD4+IL-10 cytokine level was not up-regulated in both of the chosen antigen immunized groups (Figure 3B). To support cytokine production, ED-I/I-AD molecule expression was measured on the monocytes of each group of immunized mice. As expected, the r-*Ld*ODC protein and peptide cocktail immunized mice group results showed a higher amount of CD14+ED-I/I-AD expression (*p* ≤ 0.05) on monocytes/macrophages than the placebo and adjuvant group (Figure 4A).

### 3.5. Measurement of Parasite Load in Macrophage

Cultured PBMCs from splenocytes (1 × 10^6^ cells/well, 6-well plate, Nunc, USA) of immunized mice were challenged by CFSE-stained (1 µg, BioLegend, CA, USA) *Leishmania* parasites in a 1:10 ratio. It was observed that the *Ld*ODC protein as well as the cocktail of peptides immunized group showed a significantly lower burden of parasites (2.5 times lesser, *p*-value ≤ 0.05) in the gated macrophages than the placebo and adjuvant group (Figure 4B). The SLA immunized groups were used as positive control and showed lower parasite burden in macrophages (Figure 4B). The mean fluorescent intensity (MFI) of each acquired group was analyzed on the cellquest software of BD Bioscience.

## 4. Discussion

Numerous leishmanial antigens have been designed and validated against leishmaniasis as a potential vaccine target in the murine model. However, none of them retain their protective efficiency in human clinical trials [9,10,31]. In our previous study, the vaccine potential of the r-*Ld*ODC protein and its derived synthetic peptides has been validated in vitro against VL [16,28].

In this study, the *Ld*ODC protein and its derived HLA-DRB1-restricted epitopes (synthetic peptides) were selected as vaccine potential targets against VL. The HLA-DRB1 gene was selected as it has a higher (five-fold) expression as compared to other HLA-DR alleles [32]. Further, the previous study indicated the high cross-presentation ability of HLA-DRB1-restricted epitopes through other HLA-DR alleles [33,34]. This has been further supported by HLA cross-presentation analysis, mentioning that shortlisted epitopes presented at least 54 other class-II HLA alleles, by that broadening the target populations up to 100%. Furthermore, a 100 ns molecular dynamics analysis indicated the stable nature of *Ld*-ODC-derived peptides [16].

In the present work, the r-*Ld*ODC protein, as well as the peptide cocktail immunized mice group showed a significantly higher secretion of cytokine IFN-γ, indicating antigen immunoprophylaxis. However, a negligible secretion of IFN-γ was observed in the placebo as well as the adjuvant immunized group. This agreed with the previous study, where IFN-γ was demonstrated as a principal cytokine having a role in the activation of macrophages and T cells during visceral leishmaniasis infection [35,36]. To support the finding of the in vivo study, intracellular CD4+IFN-γ cytokine was measured in each immunized mice group. The resulting data showed that CD4+T cells produced a significantly higher amount of IFN-γ against the r-*Ld*ODC protein as well as peptide cocktail, in comparison to the placebo group. On contrary, the level of IL-10 cytokine was not upregulated significantly in immunized mice groups for both the antigens. As per the literature, the pro-inflammatory cytokines limit the infection, but anti-inflammatory cytokine (IL-10) helps with *Leishmania* survival inside host macrophages and enhances disease progression [37,38].

In a murine model study, it was reported that a chimeric *Leishmania* protein efficiently proliferates T cells which produce proinflammatory cytokines. The generated cytokines promote Th1 cell activation against *Leishmania* infection [39,40]. In the present study, it was estimated that the protein, as well as peptide cocktail, immunized mice group displayed higher proliferation of T cells. On the other hand, a non-significant proliferation of T cells was observed in adjuvants as well as placebo immunized groups. In another experiment, two-fold higher (*p* ≤ 0.05) expressions of the CD14+ED-I/AD-I molecule on monocytes were observed against the purified r-*Ld*ODC protein and peptide cocktail immunized mice groups. This indicates that the higher expression of MHC molecules potentiates its antigenicity against *Leishmania*. The CD14+ED-I receptor expression was not observed on monocytes of adjuvant as well as the placebo immunized mice group. In each experiment, the SLA immunized mice group was used as a positive control of corresponding parameters. This study supports the fact that the antigens loaded MHC Class-II molecules are crucial factors for the activation of the T cell cascade [41].

In previous studies, it has been reported that the *Leishmania* parasite is an intracellular pathogen, which proliferates and multiplies inside host macrophages for their survival [42,43]. In the present study, it was found that the cultured stained parasite burden was lower (2.5 times lesser, *p*-value ≤ 0.05) in macrophages of the protein and peptide cocktail immunized mice groups. These lower numbers of parasites inside the macrophages strengthen the antigen immunoprophylactic potential against VL. On the other side, a significantly higher parasitic burden was observed in macrophages of the placebo and adjuvants immunized groups. These findings suggest that the antigens have the potential to suppress *Leishmania* infection by the activation of protective immune responses.

In conclusion, the r-*Ld*ODC protein and its derived HLA-DRB1-restricted peptides have the potential to generate effective cytokines for the activation of Th1 immune cells and suppress the *Leishmania* infection in a murine model. The future perspective includes the validation of vaccine candidates in a humanized mice model as an immunoprophylactic target against VL.

## Figures and Tables

**Figure 1 pathogens-12-00016-f001:**
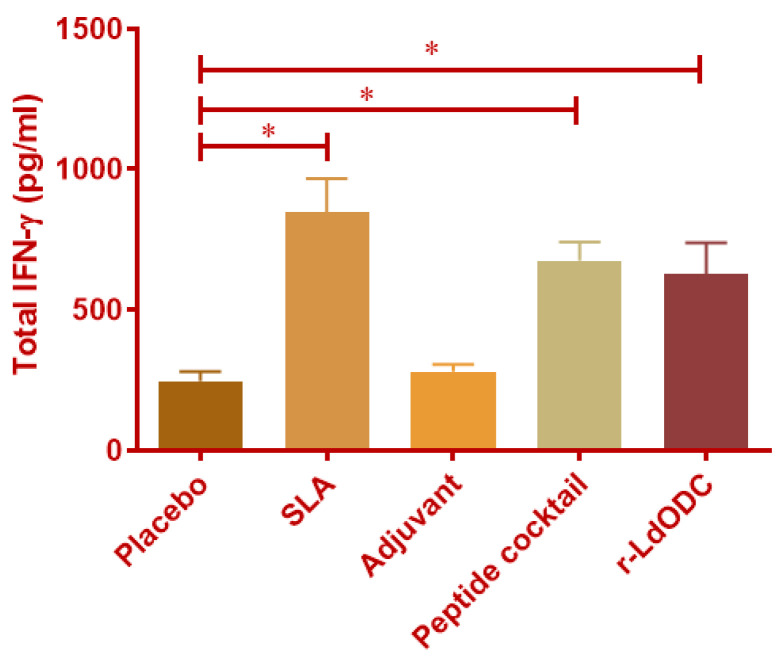
Quantitative estimation of total secretory IFN-γ in the immunized mice group. Each group of BALB/c mice (six mice in each group) was immunized with normal saline, Soluble Leishmania Antigen (SLA), Adjuvant, r-*Ld*ODC protein, or peptide cocktail. After 30 days of the last dose of vaccine, spleens were harvested and mononuclear splenocytes were isolated from each group. The mononuclear cells of splenocytes (1 × 10^6^) of immunized mice were stimulated with r-*Ld*ODC protein and peptide cocktail (10 µg/mL). The total secretory IFN-γ was quantified in cell culture supernatant of each group of mice. Analyzed data are expressed as the mean ± SEM (standard error of the mean). Asterisk bracket (*) indicates statistically significant difference (*p* ≤ 0.05) between the immunized mice group and the placebo group. The significance was assessed by one-way analysis of variance (ANOVA) with Tukey’s post hoc multiple comparison tests. Each experiment was run in triplicate, and a value of *p* ≤ 0.05 was considered for significant.

**Figure 2 pathogens-12-00016-f002:**
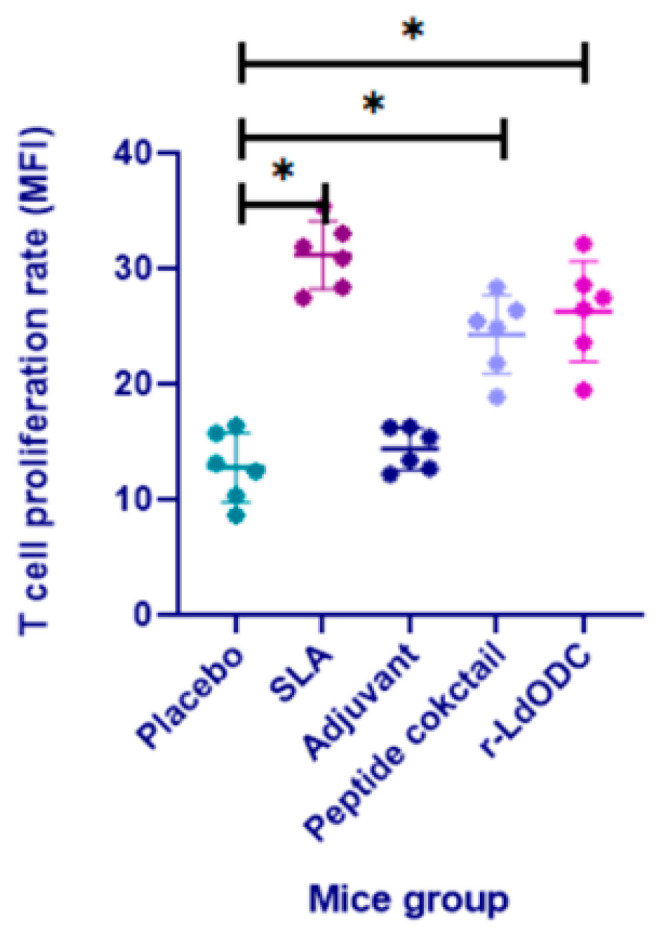
T cell proliferation analysis of CFSE-labeled PBMCs. Each group of BALB/c mice (six mice in each group) was immunized with normal saline, Soluble Leishmania Antigen (SLA), Adjuvant, r-*Ld*ODC protein, or peptide cocktail. After 30 days of the last dose of vaccine, mononuclear splenocytes were isolated from each group. The supernatant of cultured cells (1 × 10^6^) was stained separately by CFSE and added back into the same well culture plate. r-*Ld*ODC protein and peptide cocktail stimulated cells were incubated for 96 h and after incubation, T cell proliferation was measured in each group of mice. Analyzed data are expressed as the mean ± SEM (standard error of the mean). Asterisk bracket (*) indicates statistically significant difference (*p* ≤ 0.05) between the immunized mice group and the placebo group. The significance was assessed by one-way analysis of variance (ANOVA) with Tukey’s post hoc multiple comparison tests. Each experiment was run in triplicate, and a value of *p* ≤ 0.05 was considered for significant.

**Figure 3 pathogens-12-00016-f003:**
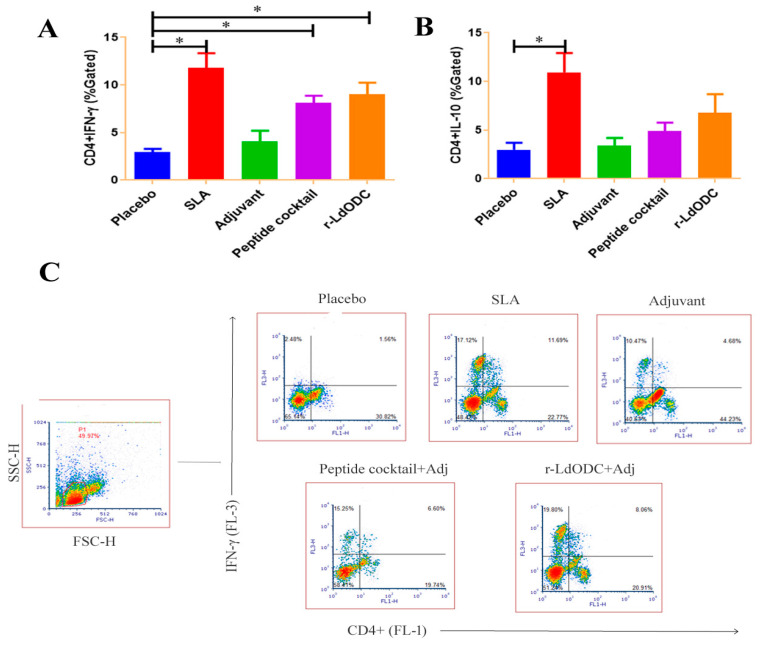
Evaluation of intracellular cytokines production in the immunized mice group. (**A**) Measurement of CD4+IFN-γ in each immunized group of mice. (**B**) Quantification of CD4+IL-10 in each immunized group of mice. (**C**) A FACS representative plot showing CD4+IFN-γ production in different immunized mice groups against normal saline, SLA, adjuvant, peptides cocktail, and recombinant protein. Each group of BALB/c mice (six mice in each group) was immunized with normal saline, Soluble Leishmania Antigen (SLA), Adjuvant, r-*Ld*ODC protein, or peptide cocktail. After 30 days past the last dose of vaccine, spleens were harvested. The mononuclear cells of splenocytes (1 × 10^6^) of immunized mice were stimulated with r-*Ld*ODC protein and peptide cocktail (10 µg/mL). Analyzed data are expressed as the mean ± SEM (standard error of the mean). Asterisk bracket (*) indicates statistically significant difference (*p* ≤ 0.05) between the immunized mice group and the placebo group. The significance was assessed by one-way analysis of variance (ANOVA) with Tukey’s post hoc multiple comparison tests. Each experiment was run in triplicate, and a value of *p* ≤ 0.05 was considered for significant.

**Figure 4 pathogens-12-00016-f004:**
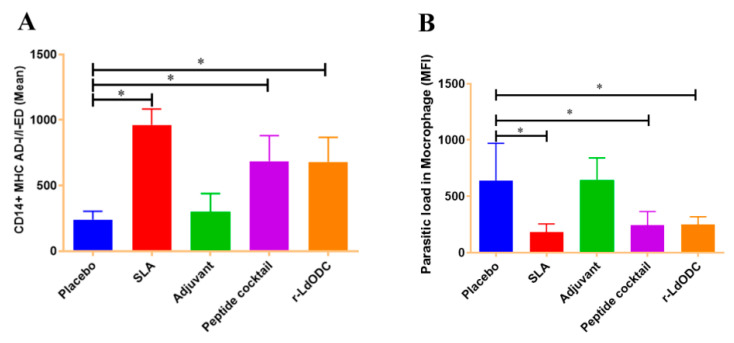
Determination of AD-I/I-ED molecules expression on CD14+ monocyte/macrophages; parasitic load in macrophages. Each group of BALB/c mice (six mice in each group) was immunized with normal saline, Soluble Leishmania Antigen (SLA), Adjuvant, r-*Ld*ODC protein, or peptide cocktail. After 30 days past the last dose of vaccine, spleens were harvested. (**A**) Measurement of AD-I/I-ED molecules expression on CD14+ monocyte/macrophages in each group where mononuclear cells of splenocytes (1 × 10^6^) of immunized mice were stimulated with r-*Ld*ODC protein and peptide cocktail (10 µg/mL). (**B**) Estimation of the parasitic load in each group of mice where the stimulated mononuclear cells of splenocytes of immunized mice were challenged by CFSE-stained *Leishmania* parasites in a 1:10 ratio. Analyzed data are expressed as the mean ± SEM (standard error of the mean). Asterisk bracket (*) indicates statistically significant difference (*p* ≤ 0.05) between the immunized mice group and the placebo group. The significance was assessed by one-way analysis of variance (ANOVA) with Tukey’s post hoc multiple comparison test. Each experiment was run in triplicate, and a value of *p* ≤ 0.05 was considered for significant.

## Data Availability

Any data related with this study are available from the corresponding author on reasonable request.

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
