# Peer review of "Validating Immunomodulatory Responses of r-LdODC Protein and Its Derived HLA-DRB1 Restricted Epitopes against Visceral Leishmaniasis in BALB/c Mice"

_pathogens, 2022, doi:10.3390/pathogens12010016_

Round 1

Reviewer 1 Report

Title: Validating Immunomodulatory Responses of r-LdODC protein and Its Derived HLA-DRB1 Restricted Epitopes against Visceral Leishmaniasis

Overview: Pandey and colleges present an article whose aim of the work was to validate the r-LdODC protein and HLA-DRB1 against visceral leishmaniasis in BALc mice.

However, the manuscript writing is not appealing to the reader, it does not make clear the importance of work for the scientific community. Some paragraphs in the text do not follow a clear explanatory logic and should be revised and reorganized. The text needs a good English review.

Therefore, in the present form, I do not consider the manuscript for publication. I recommend a full revision of the paper.

  Abstract

- The term vaccination can be change for Leishmanization ?

- The text needs a good English review.

- about this HLA... how much of the population has this allele? is it significant?

- The result of parasitism is missing in the text.

Introduction:

Reference is missing in the text.

The all text needs a good review of English. Many parts are confused.

It needs a standardization of the text, for example, there are parts that say MHC receptors... another part says MHC molecules... it needs to be revised.

 Material and Methods:

2.1 – The name of the Institution must be first befora use the inittials (RMRIMS)

- what you mean with animal house ? what specie of hamster do you use? Do you not use fetal calf sérum in the medium to cultivated leishmania?

2.4. – Why use female mice ?

- How was this antigen concentration selected?

- How were MPL and CD2 produced? and commercial? which company?

- Subcutaneous the immunization... in which part of the mice body?

2.5 - How many cells were obtained from each animal?

2.6 – selected antigens ? which antigens?

How was the standardization made for choosing the concentration 10ug/ml?

2.7 - selected antigens ? which antigens?

2.9 - what method is used to count the number of parasites?

And do you test other ratio (monocyte/parasite)?

 Results:

3.1 – This part has already been described in methodology. It's a repeat.

It would be good to include the bioinformatics part of how the peptides were generated.

3.2 – This section can be added not only the IFN-g production but IL-10 too.

- How was the standardization made for choosing the concentration of the protein and the peptides cocktail? Is the same concentration used?

- The figure legend must be corrected and the acronyms that appear in the figure must be inserted, for example Adj, PC, etc.

3.3 – The figure can be separated. In this part shows only the T cell proliferation. When you put another result withou the text is confused.

There is a graphic for MHC but no text explain the result.

The figure legend must be self explanatory. So there is no information on how the experiment was done. All legend figures must be revised.

3.4 – This item should be redirected to the item that talks about IFN. It is a repetition of item 3.2.

3.5 The figure 2B must stay in this section.

Discussion:

It is necessary an english review in all text.

Reviewer 2 Report

In line 19  Leishmanization can be used instead vaccination because was demonstrated that Leishmanization is the best way to induce effective immune response.

In line 20 what percentage of the world population or endemic areas have this allele?(HLA allele)

The authors must include the results of parasitism

in line 29 MHC molecules are not considered only or strictly as receptors

in linr 31 the authors talk about antigens i think will be bettter use the words peptide or epitope.

in line 43 the  Reference is missing in the text

in line 51 the authors can use cosst effective instead comercially

the paragraph lines 51-56 is confusing

lines 67-68  the authors meant prophylactic vaccine?

The paragraph in lines 57-74 is confusing

in line 76 Trypanothione is not exclusive to the Leishmania genus.

in line 95 "in under" ? I think is "under"

in line 96 what RMRIMS means?

in line 101 What species of hamster is used in the experiment?

in topic 2.3 is protein expression and peptide synthesis , and the authors must highlight if the protein used was LPS/endotoxin free.

in topic 2,4 why the authors used female mice?

in line 134 the manufactures of the adjuvant was missing.

in line 136 the authors talks about sub-cutaneus route ,but the injection was aplied in which part of the body?

topic 2,8 thet title must be reformulated"expression against iimunized mice"

in line 187 is events instead cells.

in materials methods the bioinformatics methosds were missing.

the authors must reformulate the manuscript

Round 2

Reviewer 1 Report

The authors need to check the figures legend. It is very confused. 

All text must have an extensive english language review. Several writing errors.

Author Response

Please, find the attached response response file 

Reviewer 2 Report

in line 90 the authors should put the hamster species which is "Mesocricetus auratus". the age of the hamsters was not clear, it was not clear if they were animals aged 8-10 weeks or animals infected for a period of 8-10 weeks.

in line 120  the phrase is " in further studies" instead "in further study" please check

in line 126  the phrase is " in further in vivo studies" instead "in further in vivo study" please check

in lines 228 -229 the authors wrote "papulation "instead "population" , "safty" instead "safety" and "caverage" instead "coverage".

the authors wrote in line 229, safty of selected peptides were elucidated by bioinformatics tools, what would these bioinformatics tools/software be?

please check the english in the lgend of figure 3 , specially lines 289-290

please check legend of figure 4, is confusing ,for example the description of the group "adjuvant (r-CD-2 & MPL-A)"

the authors must check english language and style

Author Response

Please, find the attached reviewer response file

Round 3

Reviewer 2 Report

The authors answered the questions and made the changes in the manuscript.

Author Response

Thank you so much!
